# A high-resolution real-time quantification of astrocyte cytokine secretion under shear stress for investigating hydrocephalus shunt failure

Fatemeh Khodadadei [1], Allen P. Liu [2,3,4,5] & Carolyn A. Harris [1,6,7 ✉]

It has been hypothesized that physiological shear forces acting on medical devices implanted in the brain significantly accelerate the rate to device failure in patients with chronically indwelling neuroprosthetics. In hydrocephalus shunt devices, shear forces arise from cerebrospinal fluid flow. The shunt's unacceptably high failure rate is mostly due to obstruction with adherent inflammatory cells. Astrocytes are the dominant cell type bound directly to obstructing shunts, rapidly manipulating their activation via shear stress-dependent cytokine secretion. Here we developed a total internal reflection fluorescence microscopy combined with a microfluidic shear device chip (MSDC) for quantitative analysis and direct spatial-temporal mapping of secreted cytokines at the single-cell level under physiological shear stress to identify the root cause for shunt failure. Real-time secretion imaging at 1-min time intervals enabled successful detection of a significant increase of astrocyte IL-6 cytokine secretion under shear stress greater than 0.5 dyne/cm$^2$, validating our hypothesis and highlighting the importance of reducing shear stress activation of cells.

[1] Dept. of Chemical Engineering and Materials Science, Wayne State University, Detroit, MI, USA. [2] Dept. of Mechanical Engineering, University of Michigan, Ann Arbor, MI, USA. [3] Dept. of Biomedical Engineering, University of Michigan, Ann Arbor, MI, USA. [4] Dept. of Biophysics, University of Michigan, Ann Arbor, MI, USA. [5] Dept. of Cellular and Molecular Biology Program, University of Michigan, Ann Arbor, MI, USA. [6] Dept. of Biomedical Engineering, Wayne State University, Detroit, MI, USA. [7] Dept. of Neurosurgery, Wayne State University, Detroit, MI, USA. ✉email: caharris@wayne.edu

Persistent motion-related shear forces of medical devices implanted in the brain trigger a rather complex cascade of foreign body reactions (FBR) when compared to the initial iatrogenic trauma, culminating in high device failure rates[1–3]. There is a striking difference with initial mechanical-insertion-induced acute injury and chronic tissue inflammation from indwelling probes. The primary mechanism is exacerbated glial response around anchored devices caused by micromotion and/or shear stress[4–9]. Brain motion-related shear forces arise from physiologic sources such as cardiac rhythm, fluctuation in respiratory pressure, interstitial fluid flux, and in the case of shunt systems used to treat hydrocephalus with high intracranial pressure, cerebrospinal fluid (CSF) flow. CSF shunt implantation is the most common treatment option for hydrocephalus, yet shunts are plagued by high failure rates: 40% in the first year, 90% in the first 10 years. Most of the CSF volume flows through the proximal holes of the shunt's ventricular catheter, i.e., holes located furthest from the tip of the shunt with less resistance to flow. Computational fluid dynamics simulations have shown that in CSF shunts, the wall shear stress at the proximal holes is greater than 0.5 dyne/cm$^2$ [10–12]. This fact increases the shear stress at the proximal segment and is a key driver of a dense glial scar formation around devices causing failure via obstruction[11,13,14]. Obstruction with adherent inflammatory cells is the most common cause of CSF shunt failure. Recent long-term in vivo data collected in our lab indicate that inflammatory astrocytes are ubiquitous on all shunts; they make up more than 21% of cells bound to obstructed shunts, and of the occluded masses blocking ventricular catheter holes, a vast majority of the cells are astrocytes, and their number and reactivity peak on failed shunts. Our data corroborate other reports suggesting that inflammatory astrocyte activation on shunts is correlated to a change in flow rate through the shunt holes and indirectly, the shear rate through these holes[10,15–17]. Astrocytes have an increased attachment propensity in vitro with increasing flow-induced shear stress. We have also observed astrocyte markers in obstructive masses to be co-localized with proliferative markers, indicating that astrocytes are active on the shunt surface: they produce inflammatory cytokine IL-6 and proliferate.

Astrocytes directly sense mechanical stress through activation of cell membrane flow-sensors, and translate it into cellular signaling, such as rapid increase in cytokine secretion[5,18]. Cytokine secretion creates spatially and temporally varying concentration profiles in the extracellular environment to communicate, activate, and recruit other inflammatory cells, emphasizing the dominant role of cytokines in orchestrating the dynamic crosstalk among cells. Thus, it is reasonable to hypothesize that along the shunt/CSF interface, shear stress-activated astrocytes secrete cytokines which in turn cause heightened activation and proliferation of other astrocytes and eventually accelerates glial scar formation on the device surface leading to high shunt failure rates. Cytokine quantification is the most important measurable outcome for inflammation and a starting point for thorough investigation of device failure. Fortunately, unlike shear stress in other areas of the brain, we can control shear stress on the shunt surface by simple modifications of shunt design. Therefore, there is a clear need to investigate these relationships further to reduce astrocyte activation for significantly improved device design for the next-generation medical devices.

Several groups have reported population analysis of cytokine secretion from single cells by using antibody-based immunoassay applications[19,20]. While these high throughput and/or quantitative approaches are effective, they encounter several inherent drawbacks such as intensive wash steps, which causes a delay between secretion and detection. Therefore, these approaches cannot currently offer either a time interval of shorter than a few hours nor simultaneous real-time observation of a second cellular variable (e.g., cell physiological states, intercellular adhesion molecule expression) at the time of cytokine secretion. To successfully address this issue in fluorescence immunoassay, Ohara and colleagues have taken advantage of near-field excitation in total internal reflection fluorescence microscopy (TIR-FM)[21–23]. In these studies, target proteins secreted from single cells were quantified by detecting formation of sandwich immunocomplexes on the device surface. The anti-cytokine capture antibody immobilized on the surface immediately captures the cytokine secreted from single cells, which enables TIR-FM to operate in situ. A similar approach with a label-free technique based on nanoplasmonic imaging has been established[24,25]. However, sandwich fluorescence immunoassay is a more sensitive and specific approach for small molecules like cytokines than the plasmonic approach, since the plasmonic signal is proportional to the molecular weight of the binding molecule.

In this study, we developed a unique and feasible platform by combining a microfluidic shear device chip (MSDC) with TIR-FM to better understand shear stress-dependent inflammatory astrocyte activation, through real-time quantitative analysis and direct spatial-temporal mapping of secreted cytokines upon shear stress at the single-cell level. Single-cell methods provide unprecedented detailed information about dynamic behavior of complex spatiotemporal distributions of cytokines as well as transient and rare cellular states which complement bulk assays. Here, by using our TIR-FM/MSDC biosensor we examined the correlations between physiological shear stress greater than 0.5 dyne/cm$^2$ applied on single astrocytes and the resulting IL-6 cytokine secretion to answer our question of whether shear stress significantly accelerates astrocyte activation on the CSF shunt surface. The answer will dramatically change our perspective of treatment of hydrocephalus, and sheds light on other chronically indwelling probes in the brain.

## Results

**Generation of a TIR-FM/MSDC biosensor for reproducing the in vivo brain physiological response along the shunt hole/CSF interface**. Along the shunt hole/CSF interface, astrocytes are simultaneously exposed to fluid flow shear stress and TNF-α/IL-1β cytokine stimulation secreted from activated microglia/macrophages on the shunt surface[26,27]. Based on reports, shear stress has less effect on microglia/macrophages, but a direct stimulation effect on astrocytes[4,5]. Co-stimulation with TNF-α and IL-1β induces a proliferative reactive astrocyte phenotype called A2, that secrete IL-6 cytokines[28–30]. IL-6 activates astrocyte proliferation by a positive feed-forward loop, further activating local astrocytes implicated in the induction of glial scar formation (Fig. 1a). The unique TIR-FM/MSDC biosensor effectively reproduces the brain tissue response by applying physiological shear stress on single astrocyte cells and providing quantitative data on spatial secretion of cytokines in real-time by detecting formation of cytokine sandwich immunocomplex immediately following astrocyte secretion. The anti-cytokine capture antibody immobilized on the microchannel surface, onto which secreted cytokine and fluorescently labeled detection antibody is bound, form the sandwich immunocomplex (Fig. 1b). In a microfluidic device, physiological shear stress can be created by fluid flow induction at a predetermined rate, in the presence of TNF-α/IL-1β to determine the influence of shear stress on specific astrocyte phenotypes. In this study, four microfluidic channels with different widths were designed to provide a multiplex culture platform to measure four different shear stress stimulations on astrocytes[31,32] (Fig. 2a). COMSOL simulations were conducted to demonstrate that the microchannels experience a uniform shear

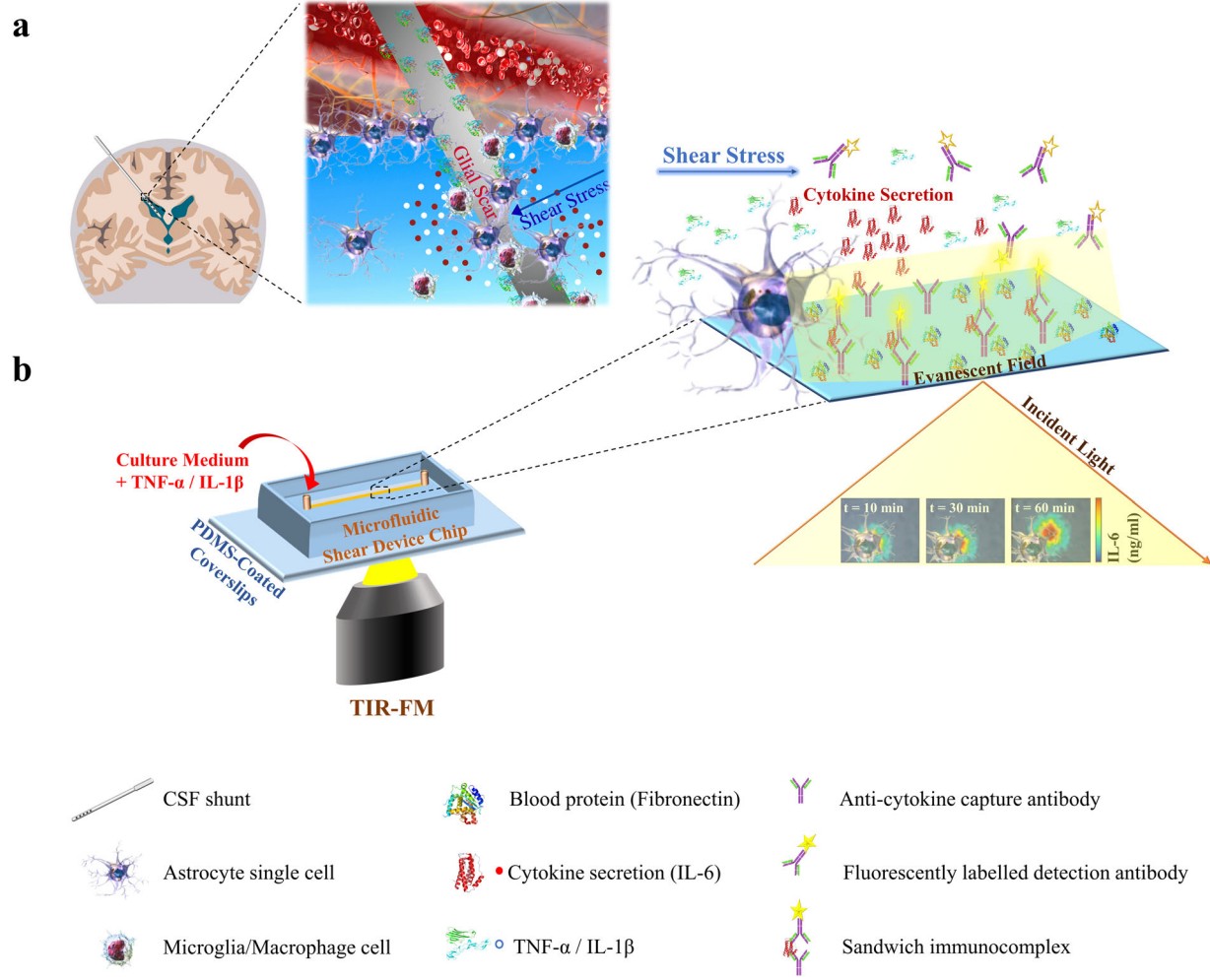

**Fig. 1 Real-time spatial monitoring of live single-cell cytokine secretion assay platform for reproducing the physiological response along the shunt hole/CSF interface.** Schematic of the **a** inflammatory FBR along the shunt hole/CSF interface **b** TIR-FM/MSDC biosensor: by taking advantage of near-field excitation in TIR-FM, target cytokines in each MSDC microchannel are quantified by detecting formation of cytokine sandwich immunocomplex immediately following single-cell secretion. The anti-cytokine capture antibody immobilized on the microchannel surface, onto which secreted cytokine and fluorescently labeled detection antibody is bound, form the sandwich immunocomplex. The detection strategy offers an advantage in its non-invasive monitoring by immobilizing and labeling target molecules in the extracellular space.

stress distribution, and thus the astrocytes are subject to a uniform shear stress (Fig. 2b). Given that computational fluid dynamics simulations have shown that in CSF shunts, the wall shear stress at the proximal holes (holes located furthest from the tip) is larger than 0.5 dyne/cm$^2$, in this study high uniform wall shear stress of more than 0.5 dyne/cm$^2$ across the microchannel's surface is maintained. The control as low shear stress is less than 0.05 dyne/cm$^2$ wall shear stress across the microchannel's surface, which is less than one tenth of the wall shear stress that induced cytokine secretion from astrocytes.

We know that the initial phase of the FBR is blood-device interactions, which occurs immediately upon implantation caused by vasculature or blood–brain barrier (BBB) disruption, resulting in the influx of serum proteins and their nonspecific adsorption to the device surface[33]. Thus, in this study for an appropriate representation of the in vivo occurrence of inflammatory response on silicone CSF shunts, polydimethylsiloxane (PDMS) coated glass substrates were selected with nonspecific adsorptions of blood proteins, such as fibronectin (FN).

**Performance evaluation of the TIR-FM/MSDC biosensor.** Experimental validation of the TIR-FM/MSDC platform was

carried out by introducing different concentrations of cytokine into microchannels, mimicking cytokine secretion from a single-cell, and quantifying the amount of fluorescence signals. Fluorescence signals increased immediately after introduction of IL-6, indicating that the introduced IL-6 was instantly captured by antibody in the microchannel. Fluorescence increases with an increase in IL-6 concentration on both glass and PDMS substrates (Fig. 3a), indicative of a successful platform. The capture ratio of the TIR-FM/MSDC biosensor depends upon variables such as the height of the cytokine release point (which determines the probability of the cytokine encountering the capture antibody). Therefore, the assay platform developed in this study is proficient for in situ detection of the onset of cytokine secretion from single cells at high temporal resolution in a sensitive and selective manner, while also providing semi-quantitative data on secreted cytokines (Fig. 3b).

**TIR-FM/MSDC biosensor for shear-cytokine activation of astrocytes.** Along the shunt hole/CSF interface, astrocytes are simultaneously exposed to fluid flow shear stress and TNF-α/IL-1β cytokine stimulation (shear-cytokine activation). To answer our question of whether under physiological conditions shear

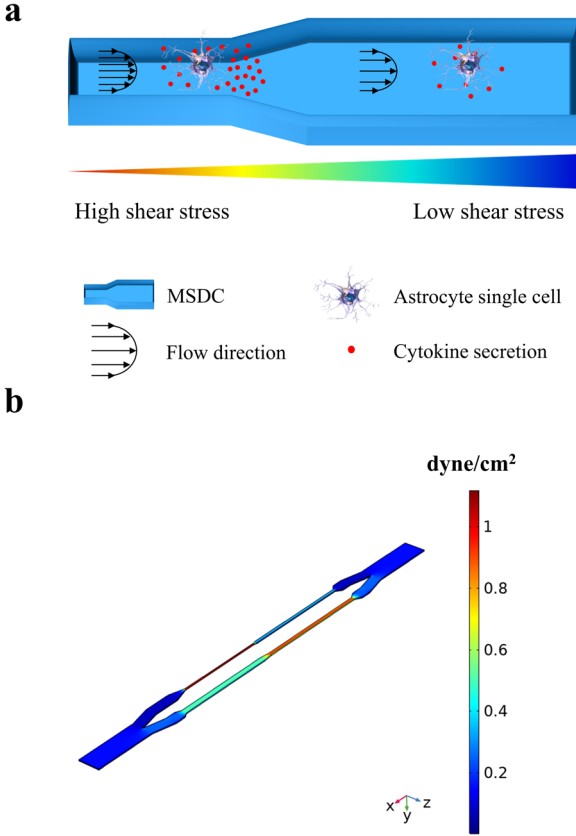

**Fig. 2 Multiplex high-throughput microfluidic device for uniform shear stress stimulation of cells. a** Schematic of the MSDC design with different widths to provide a multiplex culture platform to measure different shear stress stimulations on cells. The microchannel with a smaller width and a higher fluid flow shear stress presents higher cytokine secretion from cells. **b** COMSOL simulations of flow in the microchannels predicted relatively uniform wall shear stress values across cell adhesion areas for all TIR-FM analysis. A flow rate of about 10 µl/min generates more than 0.5 dyne/cm$^2$ of uniform wall shear stress across the widest rectangular microchannel's surface of 1000 µm width.

stress significantly accelerates astrocyte activation on the CSF shunt surface, the TIR-FM/MSDC biosensor was employed. Time- and space-resolved observation under simultaneous fluid flow shear stress and TNF-α/IL-1β cytokine activation of single human astrocytes revealed a significant increase in IL-6 secretion under high shear stress compared to low shear stress controls (Fig. 4). Initially ($t = 10$ min) when cytokine stimulation of astrocytes has not started, shear only stimulated astrocytes on glass substrates for both low and high shear stress had no IL-6 secretion. However, as time increases ($t > 30$ min) astrocytes under high shear-cytokine activation resulted in significantly ($p = 0.0015$) higher levels of IL-6 secretion compared to low shear-cytokine activation (Fig. 4a, b). Under low shear stress, IL-6 multidirectional diffusion in space around the cell is detected. However, under high shear stress, IL-6 distribution in space away from the cell is observed (Fig. 4a, c). The rising curve of IL-6 secretion from shear-cytokine activated astrocytes, indicative of an increase in IL-6 concentration, occurred as a concave-down function, suggesting IL-6 was secreted in a burst release pattern rather than continuous secretion of IL-6, based on reports[21] (Fig. 4b). Real-time monitoring of cells over time ensured that astrocytes were not detached from the surface, under high shear stress. Via this unique assay platform, simultaneous dynamic cellular states for live single cells could also be observed such as a change in astrocyte phenotype from a resting to a reactive form under shear-cytokine activation, characterized by horizontally swollen bodies (Fig. 4d). Swelling of reactive astrocytes was quantified as the increase in cross-sectional area relative to the resting astrocytes[28,34]. On PDMS substrates, similar results for shear stress were observed, with high and low shear created a significant difference in the IL-6 secretion after 30 min of activation ($p < 0.031$). However, astrocytes on hydrophobic PDMS substrates, initially displayed higher activation of IL-6 secretion compared to hydrophilic glass substrates under high shear stress. Together, these data provide evidence that astrocytes directly sense the mechanical stress and translate it into chemical messages of rapid increase in IL-6 cytokine secretion.

Shear-cytokine activation of astrocytes was simultaneously performed using the ELISpot assay on a second prepared MSDC[35]. The same procedure is used as TIR-FM experiments, with the difference of detecting signals for IL-6 secretion from a

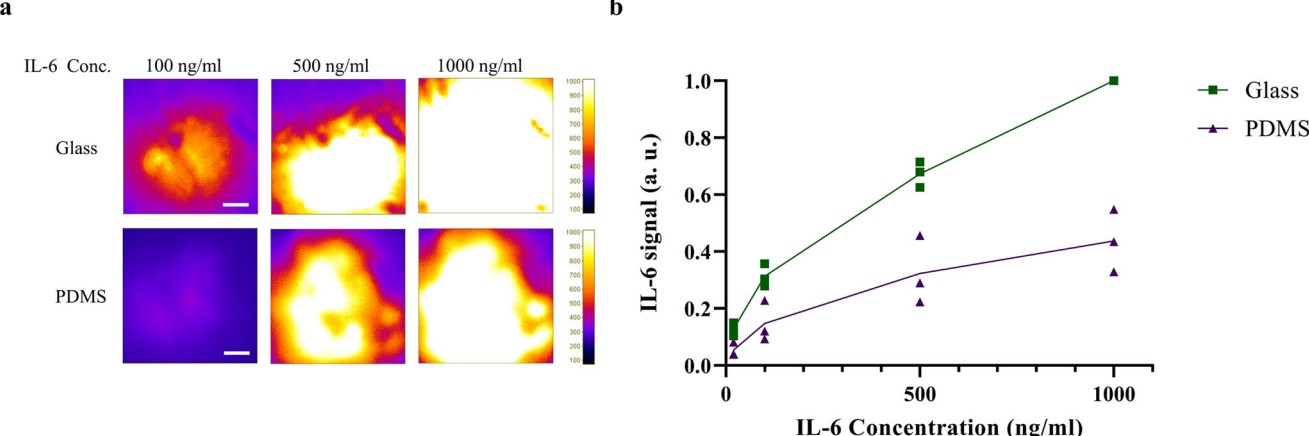

**Fig. 3 Standard curve for performance evaluation of the TIR-FM/MSDC biosensor. a** Representative images of developed fluorescence signals obtained after introducing different concentrations of IL-6. An increase in fluorescence signal was observed after an increase in IL-6 concentration. **b** The average intensity of the IL-6 quantified values from **a**. Glass substrates (green) were more efficient at binding IL-6 compared to PDMS substrates (purple). The results are represented as individual data points in the plots and the solid line denotes the average value. Data were normalized via dividing by the maximum value to bring all the measured values in the dataset to the same metric scale. Independent experiments were performed on three different devices ($n = 3$) per substrate. Scale bars represent 10 µm.

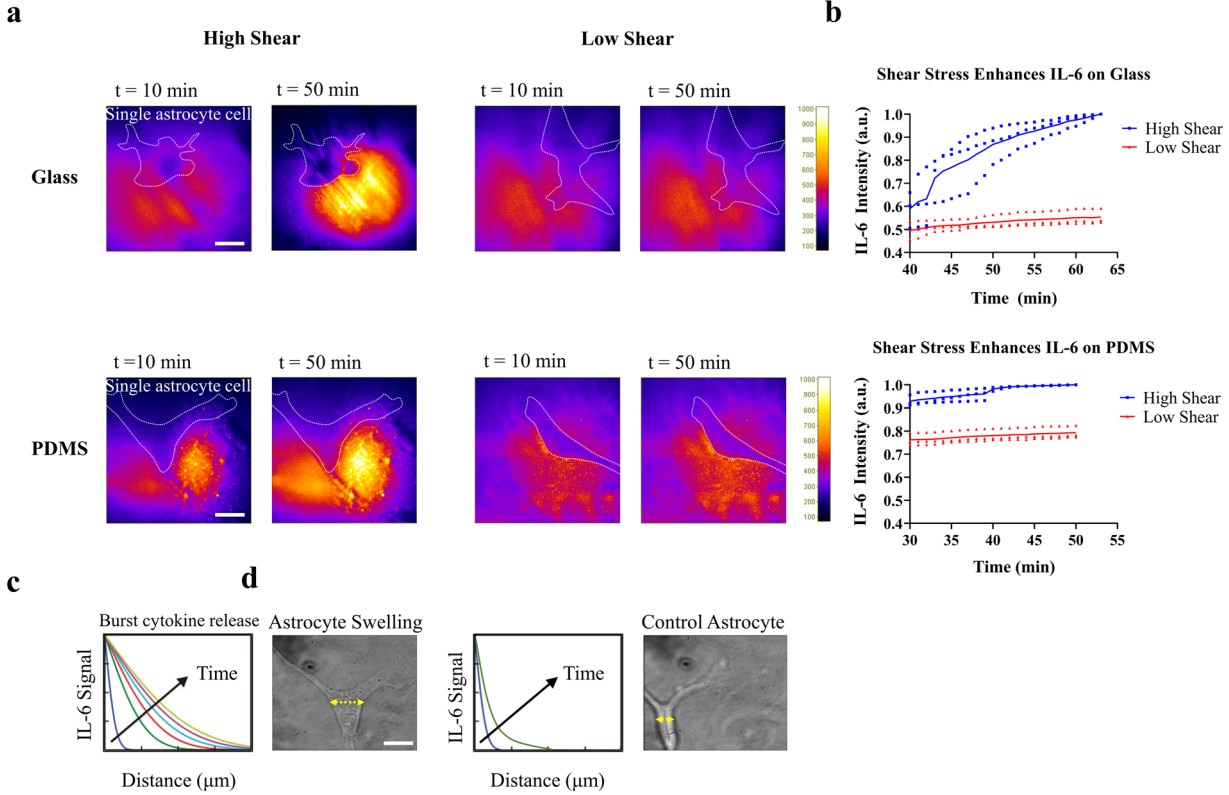

**Fig. 4 Spatial-temporal mapping of secreted cytokines from single astrocyte cells upon shear stress on different substrates. a** Quantitative spatial-temporal resolution mapping of IL-6 concentration under low and high shear stress at time 10 and 50 min, on glass and PDMS substrates. **b** Time course of the average intensity of IL-6 signals under low and high shear. The results are represented as individual data points in the plots and the solid line denotes the average value. Data were normalized via dividing by the maximum value to bring all the measured values in the dataset to the same metric scale. Independent experiments were performed on three different devices ($n = 3$) per substrate. **c** Representative measures depicting spatial-temporal mapping of IL-6 signal dependent on distance from astrocytes for a duration of 50 min. **d** Brightfield image of resting and high shear-cytokine activated single astrocyte cell, characterized by horizontally swollen bodies. The cross-sectional area was calculated using a line drawn to measure the diameter of astrocyte boundary. Single astrocyte cells were identified and localized in individual image frames. Scale bars represent 10 μm.

population of astrocytes attached on the surface, instead of single astrocytes (Fig. 5a). In accordance with our TIR-FM data, ELISpot data confirm that under high shear stress, despite less cell attachment on both glass and PDMS substrates compared to low shear stress, a significant ($p = 0.0056$ and $p = 0.0005$, respectively) increase in IL-6 secretion was detected (Fig. 5b).

**Real-time shear-cytokine activation of astrocytes on FN-coated substrates**. The unique real-time feature of our platform enables early cytokine activation detection of cells, for an accurate comprehension of earlier events in interfacing technology, leading to cumulative enhancements. Time- and space-resolved observation under simultaneous fluid flow shear stress and TNF-α/IL-1β cytokine stimulation of single human astrocytes on FN-coated PDMS/glass substrates strikingly revealed astrocyte activation and high IL-6 signals at time zero, indicating that astrocytes had already been extremely activated, and had already released IL-6 on FN-coated substrates before shear-cytokine activation. However, the identical burst release pattern of IL-6 secretion is also observed under high shear stress for FN-coated substrates in time. The degree of astrocyte activation differs significantly on FN-coated hydrophobic/hydrophilic surfaces (Fig. 6a). ELISpot data confirm the results obtained from the TIR-FM/MSDC biosensor. (Fig. 6b).

**Discussion**
In this study simultaneous shear-cytokine activation of astrocytes was performed, showing significantly higher levels of IL-6

secretion under higher shear stress compared to lower shear stress. Initially, shear-only stimulated astrocytes resulted in no IL-6 secretion on glass substrates for both low and high shear stress, indicating that shear stress alone does no induce astrocyte activation. PDMS substrates show a very minimal increase of IL-6 signal at time zero. However, as time increases astrocytes under high shear stress resulted in significantly higher levels of IL-6 secretion on both substrates. The simultaneous effect of shear-cytokine activation had a strong effect on IL-6 secretion, higher than only cytokine (TNF-α/IL-1β) activation observed for low shear stress. These findings are consistent with other reports[36–38]. Our results verify that along the shunts proximal hole (hole located furthest from the tip) with the highest shear stress, astrocytes are significantly activated to secrete IL-6. Since IL-6 activates astrocyte proliferation by a positive feed-forward loop, local astrocytes are also significantly activated and proliferate to form the glial scar on the shunts proximal hole[39,40], leading to high shunt device failure rates. Moreover, recent long-term in vivo data collected in our lab indicate astrocyte markers in obstructive masses on shunts to be co-localized with proliferative markers, indicating that astrocytes are active on the shunt surface: they produce inflammatory cytokine IL-6 and proliferate.

ELISpot data confirm that under higher shear stress, despite less cell attachment to the surface, a significant increase in IL-6 secretion is detected. This is strongly in accordance with other reports, which indicate a necessity for control of the degree of inflammatory cell activation for considerable improvement of device performance within the brain, in contrast to the general

**a**

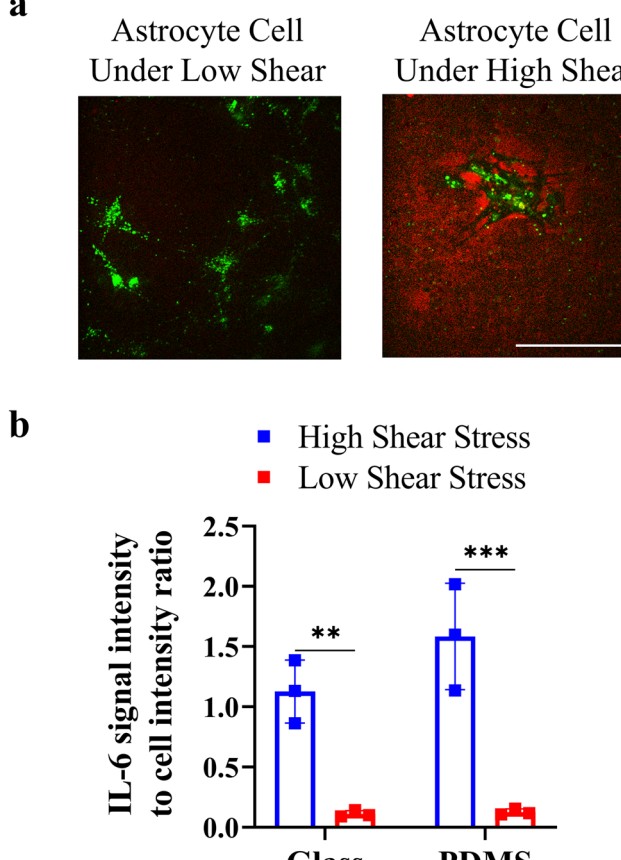

Astrocyte Cell Under Low Shear     Astrocyte Cell Under High Shear

**b**

■ High Shear Stress
■ Low Shear Stress

**Fig. 5 ELISpot population analysis of astrocyte under low and high shear stress on different substrates. a** Representative fluorescence image for glass substrate qualitatively depicting IL-6 signal (red) distribution merged with astrocyte signal (green), and **b** IL-6 signal intensity to cell signal intensity ratio. After shear-cytokine activation of astrocyte, astrocytes were stained with DiO lipophilic membrane labeling dyes. Each data point denotes an average value for $n = 5$ cells measured. Intensity values were normalized for the number of cells present. Independent experiments were performed on three different devices ($n = 3$) per substrate. The results are represented as mean ± SD in the plots. Scale bar represents 100 μm. The $p$ values ** and *** denote $p = 0.0056$ and $p = 0.0005$.

implant failure of reduced cell adhesion on the device surface in vivo[3,41–43]. Our results present a proof of concept that to have maximal impact, strategies should implement a focus on attenuating the initial inflammatory cell activation instead of only focusing on reducing cell adhesion on the device surface. Such strategies include decreasing shear activation as a primary cause of device failure, and directly antagonizing the accumulation of pro-inflammatory cytokines via targeted therapeutic for TNF-α and IL-6.

Single astrocyte cells were identified and localized in individual image frames at least 100 μm apart from other astrocytes. Since the distance over which cytokine-mediated communication happens effectively is far less than 100 μm, and cytokine signaling are limited to small distances of only a few cell diameters[44–46], substantiates that the IL-6 secretion detected is only from the identified single astrocyte cell. To observe simultaneous dynamic cellular states under shear-cytokine activation, future work will expand on the degree of cell reactivity by quantifying astrocyte processes directionality and bifurcation number.

One advantage of our platform is the direct effect of surface property on the degree of astrocyte activation under physiological

shear-cytokine activation. Under conditions of high shear stress, astrocytes on hydrophobic PDMS substrates resulted in higher IL-6 secretion compared to hydrophilic glass substrates, possibly due to cell spreading and contact between the attached cells and ligand adsorption on different surfaces. Furthermore, ELISpot data confirm that under low shear stress, there is nearly no change in the degree of astrocyte activation on hydrophobic/hydrophilic surfaces, indicating the high impact of shear activation on implant failure.

Considering that fluorescence signals for IL-6 detection on PDMS substrates were lower than on glass substrates for the generation of the standard curve (Fig. 3), we expect data are underestimating the degree of astrocyte activation on PDMS (Fig. 4). That is, under high shear stress, we expect the addition of cells to generate further astrocyte activation on PDMS substrates.

The unique real-time feature of our platform enables early cytokine activation detection of cells, to measure earlier events of disease progression. To highlight the importance, we observed that FN-coated PDMS/glass substrates result in high levels of IL-6 signals, at time zero, indicating that FN alone extremely activates astrocytes to secrete IL-6. Such information is not obtainable by other traditional techniques. FN participates in converting quiescent astrocytes to a proliferating A2 phenotype in glial scar[47,48]. These findings are consistent with other reports, indicating FN increases cell activation, which leads to differences in protein expressions[49,50]. Under high shear stress the burst release pattern of IL-6 secretion is observed for FN-coated substrates, once more indicating the impact of shear stress activation. ELISpot data confirm the results obtained from the TIR-FM/MSDC biosensor. Also, of interest is the degree of astrocyte activation as it differs significantly on FN-coated hydrophobic/hydrophilic surfaces. Since PDMS substrates without FN already show a minimal increase of IL-6 signal and astrocyte activation at time zero, PDMS substrates with FN show less effect on astrocyte activation compared to glass substrates with FN.

Therefore, since an arm of shunt obstruction is arguably originated from single astrocytes shed into the CSF, we can assume that shear stress activates these astrocytes to secrete cytokines. This would induce exaggerated proliferation in astrocytes as they bind to the FN adsorbed on shunt surfaces[51]. Even so, if we assume shunt obstruction also originates from astrocytes migrating into the shunt holes from the parenchyma or ventricular wall, these cells would be exposed to high shear stress and similarly release IL-6 for exaggerated proliferation[10,15,16].

Another advantage of real-time cytokine secretion dynamic in space is elucidating the chronological relationship between intracellular events by allowing simultaneous monitoring of a second cellular variable. For example, since cell adhesion increases as well with shear stress on astrocytes via intercellular adhesion molecule-1 (ICAM-1) ligation on astrocytes, and TNF-α and IL-1β mediate ICAM-1 induction via microglia/macrophages–astrocyte interaction in CNS injury[52], the TIR-FM/MSDC biosensor will allow for simultaneous visualization of ICAM-1 expression on astrocyte cells and cytokine secretion profiling under shear stress, and their chronological relationships.

A precise selection of astrocyte phenotype leads to a precise control of astrocyte activation. In a landmark study, Barres and colleagues identified two significantly different reactive astrocyte phenotype which they called A1 and A2[28,29]. The A1 neuroinflammatory astrocytes are induced by NF-κB signaling, whereas A2 scar-forming, proliferative astrocytes are induced by STAT3-mediated signaling[28,39]. Given that glial scar borders are formed by newly proliferated, elongated astrocytes via STAT3-dependent mechanisms, studies strongly suggest that the A2 reactive astrocyte phenotype is present during glial scar formation[29,53,54]. In this study, serum-cultured human astrocytes are used which

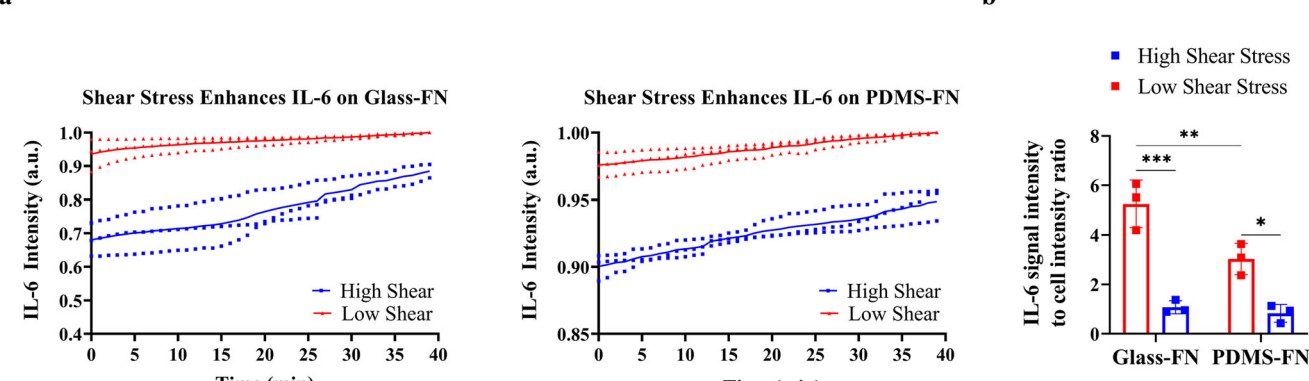

**Fig. 6 Fibronectin activation of astrocytes on different surfaces. a** Time course of the average intensity of IL-6 signals under low and high shear stress for glass and PDMS substrates. The results are represented as individual data points in the plots and the solid line denotes the average value. Data were normalized via dividing by the maximum value to bring all the measured values in the dataset to the same metric scale. Independent experiments were performed on three different devices ($n = 3$) per substrate. **b** ELISpot population analysis of astrocyte under low and high shear stress on glass and PDMS substrates characterized as: IL-6 signal intensity to cell signal intensity ratio. Each data point denotes an average value for $n = 5$ cells measured. Intensity values were normalized for the number of cells present. Independent experiments were performed on three different devices ($n = 3$) per substrate. The results are represented as mean ± SD in the plots. The $p$ values *, **, and *** denote $p = 0.0102$, $p = 0.0095$, and $p = 0.0002$.

based on reports express many of the A2 astrocyte genes[39,55], since in vivo, quiescent astrocytes contact serum upon injury and BBB disruption[56]. Additionally, IL-6 signaling pathways are enhanced in A2 astrocyte phenotypes, and STAT3 is activated by IL-6 implicated in the induction of glial scar formation on the device surface[39,57]. Co-stimulation with TNF-α and IL-1β induces A2 reactive astrocyte phenotype[30]. Interestingly, expression of TNF-α, IL-1β, and IL-6, is rapidly upregulated in the injured CNS, and are observed right at the device-tissue interface corresponding to the location of activated microglia/macrophages and astrocytes[3,27,33,58]. IL-6 promotes neuronal survival and neurite growth. These features are indications of the beneficial roles of IL-6 in repair and modulation of inflammation in the CNS. However, the overproduction of IL-6 is associated with glial scar formation, and periventricular white matter (where neuroinflammation occurs) injury due to increased CSF concentrations of IL-6[59]. Thus, a careful inflammatory balance of IL-6 is necessary for proper repair. Inhibition of both IL-6 and IL-6r by antibody neutralization reduces damage to surrounding brain parenchyma as a result of bystander effects of increased CSF cytokine levels[60].

CSF shunts removed for obstruction show occlusions to occur most often at the proximal holes (holes located furthest from the tip) with the highest flow[61]. These observations led to a suggestion that shunt geometry with a more uniform flow rate distribution among the shunt's inlet holes would reduce the obstruction occurring at the critical proximal inlet holes, thereby reducing shunt failure rates. As a result of this research, the Rivulet (Medtronic Neurosurgery) shunt was developed with a design consisting of decreasing hole diameters from the distal to proximal end[11,62]. However, shear stress will be higher in the proximal holes of these shunts. Based on fluid shear stress equation of $\tau = \mu\ du/dy$, where $\mu$ is dynamic viscosity, and $du/dy$ is the gradient of velocity in the direction perpendicular to the flow, since the gradient velocity of the decreasing hole diameters is higher, shear stress will be higher for the proximal holes. Based on our hypothesis and other reports of the correlation between increased shear stress along the shunt/CSF interface to result in increased occlusion[10,15,16,63,64], it is essential that the next-generation shunt design prototype attempts to decrease shear stress through all its holes or at best the proximal holes.

Another aspect to address is shunt size and geometry as it is relevant to the physical limitations of shunt design. Physical and surgical limitations of ventricular size, ventricular catheter environment, and ventricular catheter tip location to avoid hitting capillaries all have effect on the persistent high shunt failure rates particularly when dealing with pediatric populations. As we continue to design shunts, we will be working with neurosurgeons to accentuate lower shear stress according to the physical limitations of ventricles. For example, exploring novel ways to improve the size of the holes and inner diameter for better compatibility with pediatric patients.

Our platform provides a comprehensive picture of the real-time single-cell modes of multiplex cytokine secretion profiles in space around the cell: one directional/multidirectional, autocrine/paracrine, and continuous/concentrated burst. Cells release some cytokines in one direction to impart synaptic cell-to-cell communication and others multidirectional to establish chemokine gradients and recruit inflammatory cells. For example, to prevent shunt obstruction, other than manipulating the shunt geometry, drug therapy is also used for inhibition of cytokines and therefore inhibition of scar formation[65]. However, cytokines that are secreted in one direction may be less accessible to blocking agents and therefore more difficult to target than multidirectional cytokine secretions[66]. Therefore, spatiotemporally determining cytokine secretion profiles in response to shear forces using the TIR-FM/MSDC biosensor is critical for predictive models of immune responses to prevent cell activation. Since astrocytes express cytokines in a threshold dose- and time-dependent fashion, parameters for a controlled delivery could also be determined, specifically, the onset point, the dosage, and the duration. Ultimately, our platform allows for improved device design together with drug therapies for the better-quality of next-generation medical devices. For example, since IL-6 production is considered a downstream sequence related to TNF-α/IL-1β stimulation of cells, FDA-approved neutralizing monoclonal antibody therapies that inhibit human TNF-α, IL-1β, and the IL-1 receptor have been developed to decrease their activity[28,65,67].

For future work, it would be quite interesting to demonstrate the effects of a MSDC composed of a series of microchannels with varying diameter along their length (i.e., narrowing at various points along their length) to induce fluid flow shear stress[68]. Also, to maintain the environmental conditions observed in hydrocephalus, astrocyte activation under experimentally manipulated conditions of CSF pressure, pulsation rate, and flow rates over

long periods of time serve as a potential area for future work. However, pulsation rate does not influence overall cell attachment in an acute, 20-hour period[15].

## Methods

**Reagents**. Capture and detection antibodies used for sandwich immunoassays for human IL-6 monoclonal antibody (clone 6708; MAB206) and human IL-6 biotinylated affinity purified polyclonal antibody (BAF206) were purchased from R&D Systems. CF660R streptavidin was purchased from Biotium (29040; Hayward, CA, USA). Human astrocyte cells were purchased from ScienCell Research Laboratories (Catalog no. 1800) commercial cell lines. FN from human plasma was purchased from sigma. Sylgard-184 elastomer and curing agents were purchased from Dow Corning. Human tumor necrosis factor-α (hTNF-α) no. 8902 and human interleukin-1β (hIL-1β) no. 8900 were purchased from Cell Signaling Technology.

**Microfluidic shear device chip (MSDC)**. Standard soft lithography techniques were used to create the silicon master mold from which PDMS stamps were made. PDMS stamps were prepared by mixing Sylgard-184 elastomer and curing agents at a ratio of 10:1 (w/v), casting over the mold, and curing at 60 °C overnight. The PDMS stamp and glass coverslip were permanently bonded together after the contact surfaces between them were plasma-treated. PDMS-coated coverslips were prepared by spin-coating a layer of PDMS diluted in hexane (1:20) at 5000 rpm for 2 min[69]. About 40 μg/ml FN was added to the PDMS-coated coverslips and glass coverslips then incubated for 1 h at room temperature. A 100 μl mixture of capture antibodies (100 μg/ml) was loaded onto the PDMS-coated coverslips and glass coverslips. The surface was blocked with Pierce protein-free blocking buffers.

**TIR-FM/MSDC biosensor setup**. Astrocytes cells were prepared in a $CO_2$ incubator at 37 °C in a humidified atmosphere with 5% $CO_2$. The cells were incubated in astrocyte medium containing 2% FBS (ScienCell Instruction). To permit phenotypic maturation of astrocytes, astrocytes were used in all experiments at passage 3. Astrocytes were tested for presence of glial fibrillary acidic protein, of which all cells are positive in the culture conditions described. The day before TIR-FM experiments astrocytes cells at a low concentration were flowed into the MSDC at 0.5 μl/min for 10 min and cultured overnight. Cell culture medium was flowed at 0.5 μl/min to remove the suspension cell before stressing them. Different flow rates were applied with the detection medium containing CF-labeled detection antibody (30 nM), TNF-α (100 ng/ml) and IL-1β (2 ng/ml). The cells were monitored before and after stimulation. For imaging, the coverslip chamber containing cells were transferred to a 37 °C pre-warmed microscope stage. TIR-FM imaging was performed using a Nikon TiE-Perfect Focus System (PFS) microscope equipped with an Apochrmomat 100× objective (NA 1.49), a sCMOS camera (Flash 4.0; Hamamatsu Photonics, Japan) and a laser launch controlled by an acousto-optical tunable filter (AOTF). Image acquisition was controlled by ImageJ Micro-manager software (NIH). The 640 nm channel time-lapse image series were acquired by using 200 ms exposure at 5 s interval for 120 min. Single astrocyte cells were identified and localized in individual image frames[69,70].

**Preparation of detection medium**. Detection antibody labeled with biotin was coupled with CF-labeled streptavidin at 1:10 molar ratios at room temperature in the dark for 3 h. Unoccupied sites on streptavidin were blocked with excess dPEG4-biotin acid (10199; Quanta BioDesign, Ltd., Powell, OH, USA). Unconjugated streptavidin and dPEG4-biotin were removed by ultrafiltration (Amicon Ultra-0.5, 100 kDa; Merck Millipore, Billerica, MA, USA). The detection media contained prepared CF-labeled detection antibody (30 nM), TNF-α (100 ng/ml), and IL-1β (2 ng/ml).

**COMSOL simulations of shear stress in microchannels**. In a microfluidic device, physical shear stress can simply be regulated by fluid flow induction for a laminar flow behavior. This concept is based on the laminar flow Navier-Stokes theory, since the flow in the microfluidic channel is laminar, Newtonian flow, and incompressible. The laminar shear stress in the microfluidic channel can be estimated as:

$$\tau_{wall} = \frac{6\mu Q}{wh^2}$$

in the unit of dyne/cm² (1 Pascal = 10 dyne/cm²), μ is the fluid dynamic viscosity, $Q$ is the volumetric flow rate, $h$ is the channel height, and $w$ is channel width. Thus, the mechanical shear stress generated on the cell in the channel is proportionally relative to the flow rate. Small dimensions associated with micrometer-sized channels ensure laminar flow even at very high linear fluid rates. COMSOL simulations were conducted to demonstrate that the microchannels experience a uniform shear stress distribution.

In this study, the design of the microfluidic shear devices was motivated by the experimental objective of four microfluidic channels with different width to provide a multiplex culture platform to measure four different shear stress stimulation on astrocytes. However, to eliminate any geometry-dependent factor, the data collected in this study are only for the widest channel of 1000 μm width.

Therefore, the volumetric flow rate $Q$ changed for different shear stress stimulations.

**ELISpot**. Simultaneous ELISpot assay was performed with the same cell preparations and the same shear stress conditions as the MSDC. The chip surface was coated with antibodies, and the cytokines secreted by astrocytes were trapped on the surface around the cell. Binding of the cytokine to the specific antibodies formed fluorescent signals around the cell, which were acquired using confocal microscopy: Olympus-IX81 fluorescence microscope with spinning disk confocal scanner unit (CSU-X1; Yokogawa, Japan), EMCCD camera (iXon X3; Andor, South Windsor, CT), 60× objective (NA = 1.42).

**Statistics and Reproducibility**. GraphPad Prism 8 was applied for data analysis. To statistically define data normality the Anderson–Darling test was used. After homoscedasticity and QQ plots were created, a two-way repeat measure ANOVA was used with an alpha value set to 0.05 with sphericity assumed. Sidak's multiple comparisons test was performed to observe post hoc multiple comparisons.

For TIR-FM and ELISpot, independent experiments were performed on three different devices ($n = 3$) per substrate (glass and PDMS). For ELISpot, the number of replicates for each data point were at least five cells measured ($n = 5$), provided in the corresponding figure captions.

**Reporting Summary**. Further information on research design is available in the Nature Research Reporting Summary linked to this article.

## Data availability

All data generated/analyzed during this study are included in this published article. Supplementary Data 1 contains the source data for TIR-FM and ELISpot assays. All other relevant data that support the findings of this study are available from the corresponding author upon reasonable request. Data are currently stored locally and approved cloud services at Wayne State University.

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

## Acknowledgements

We acknowledge Jophin Joseph from University of Michigan for helping with TIR-FM imaging and Christopher Roberts from Wayne State University for helping with shear stress simulations. We acknowledge the University of Michigan for hosting this research. This work was supported by Wayne State University internal funding.

## Author contributions

F.K., C.A.H., and A.P.L. conceived the study and designed the experiments. F.K. performed the experiments and wrote the paper. All authors commented on the paper and contributed to it.

## Competing interests

The authors declare no competing interests.
