## [Peer Review File · Communications Biology]

Reviewers' comments:

Reviewer #1 (Remarks to the Author):

Overall, this is an excellent, timely, and very novel study with an important impact on the mechanisms that influence tissue obstruction of shunt catheters. The data are solid except for a minor concern about how astrocyte phenotypes are analyzed. While the text is very informative, the main concern is that much of the details presented in the Introduction and parts of the Results should be included in the Discussion. This could also shorten the manuscript somewhat. This study should definitely be published after some revisions are made to the text.

Abstract

1. "maneuvering" is a strange word to use to describe movement of interstitial fluid; something like "flux" seems more physiological.
2. The content of the abstract is heavily weighted toward introductory information, with a paucity of detailed results. For example, so much emphasis is placed on shunt obstruction that the reader may assume that the experiments involve astrocytes attached to silicone catheters. More importantly, experimental design details are missing, i.e. duration of shear stress and the procedure for producing shear stress.

Introduction

3. Overall, this Introduction is very informative but excessively long and filled with details that should be reserved for the Discussion (e.g. lines 70-115). As a result, it takes forever to get to the main objectives of the study, which are partly technical and partly neurobiological.
4. The writing style should be revised in many instances. Some examples are: (a) in line 64, the word "exasperated" should be replaced by a less-emotional word such as "heightened"; (b) in line 83, "Emphasizing the dominant role of cytokines.... is not a sentence; (c) in lines 90-91, "...intensive wash steps, 91 which causes a lag...; (d) in lines 92-93, "...observation of a cellular variables..."
5. In line 41, "triggers" should be plural.
6. In line 50, shear stress may not be the only important driver of scar formation, so "the key" should be softened.

Results

7. At times the text in this section often begins with excessive background information that does not belong in a Results section (e.g. lines 120-135) and should be moved to the Discussion section.
8. The statement in lines 154-156 should be referenced.
9. Most importantly, how do we know these are (a) astrocytes and (b) that they are single cells? It is impossible to know from the photographs in Figure 4D, and in the Methods section that follows there is no description of how the cells are obtained/harvested and identified. This is important because the statement is made that "Dynamic cellular states for live single cells could also be observed such as change in astrocyte phenotype from a resting to a reactive form under shear-cytokine activation, characterized by horizontally swelled cell bodies."; other features of resting vs reactive astrocytes, i.e. branching patterns, have not been evaluated, and the astrocyte "swelling" has not been quantified.
1. Important portions of Figures 1 and 2 are extremely small and difficult to read (e.g. Figure 4c&d).
2. Validations of the TIR-FM/MSDC biosensor (Figure 3) are excellent, with caveats for the semi-quantitative nature of the data.

Methods

3. Although astrocyte-promoting media are used, no descriptions of the source of astrocytes has been provided; thus, it is not known if they are commercial cell lines or cells derived from primary

harvesting and cell culture.

4. Likewise, no cell labeling has been provided to confirm that these cells are astrocytes. This problem is compounded because the magnification used in Figure 4 does not permit visualization of the branching pattern of the cells, which could be used to distinguish astrocytes.

5. On line 479, the Author Instructions call for this heading to be “titled “Statistics and Reproducibility””.

Discussion

6. The point about IL-6 activation of astrocyte proliferation is very well taken (and should be referenced), but some mention should be made of the caveat that these relatively short-term experiments could not determine if astrocyte proliferation occurred in this in vitro system.

7. The two sentences in lines 305-308 are awkward; “In contrast to the general implant failure...” does not have a verb and thus is not a complete sentence.

8. Likewise, “Possibly, due to cell spreading...” (line 335) is not a complete sentence.

9. The statements made in lines 352-355 on the effects of IL-6 secretion seem contradictory and should be clarified.

10. In line 382, what does “density” mean?

11. The important statement about drug interventions in lines 398-399 should be referenced. In fact, it would be worthwhile to mention some of the experimental studies that have attempted to reduce neuroinflammation and astrocytosis.

12. It might be worth mentioning the “bystander effects” of increased CSF cytokine levels on the brain parenchyma, especially the periventricular white matter where neuroinflammation is known to occur.

Reviewer #2 (Remarks to the Author):

Summary: Shunt malfunction is a common problem in patients with hydrocephalus, with occlusion representing the most common cause. Here, the authors hypothesize that the effects of shear stresses at the proximal end of the shunt catheter are strongly responsible for astrocyte activation via the secretion of the pro-inflammatory cytokine, IL-6. The authors combine a microfluidic chip to simulate shear forces and surface characteristics with fluorescence-based imaging and finite element models to interrogate the role of shear-stresses in IL-6 secretion. Overall, the study is clear, convincing and well-written. It represents an important contribution to the fields of both basic biology and device design. I recommend publication with minor revisions, with specific questions outlined below.

1) In the study, 4 different microchannels with varying diameters simulate shear stresses in a high throughput manner. As a confirmation of the hypothesis, it would be useful to demonstrate the effects of microchannels with varying diameters along their length (i.e, narrowing at various points along their length) to induce shear stress concentrations.

2) While the authors make a convincing argument that reducing shear-stresses at the proximal end of the shunt would reduce fibrosis and occlusion, there are practical aspects to shunt design, such as sizes when dealing with pediatric populations. These are particularly relevant when considering that shunt failure rates are highest in infants and younger patients. The study would be boosted if the author’s discussed the implications of their findings in terms of sizing of shunt size and geometry, and their compatibility with pediatric populations.

3) CSF flow is highly intermittent. How does pulsing the flow rate affect IL-6 activation?

Response to Referees

- Text in black: reviewer's comments.
 - Text in blue: authors response.
 - Text in green: changes incorporated in the manuscript.
-

Comments of Reviewer # 1 and response by the authors:

Overall, this is an excellent, timely, and very novel study with an important impact on the mechanisms that influence tissue obstruction of shunt catheters. The data are solid except for a minor concern about how astrocyte phenotypes are analyzed. While the text is very informative, the main concern is that much of the details presented in the Introduction and parts of the Results should be included in the Discussion. This could also shorten the manuscript somewhat. This study should definitely be published after some revisions are made to the text.

Thank you so much for your interest and constructive feedback on our manuscript. We have addressed your comments below and feel that the work is greatly improved as a result of your input.

Abstract

1. "maneuvering" is a strange word to use to describe movement of interstitial fluid; something like "flux" seems more physiological.
 - ✓ We agree, the word "maneuvering" has been replaced by "flux":

Line 36: "...interstitial fluid flux...".
2. The content of the abstract is heavily weighted toward introductory information, with a paucity of detailed results. For example, so much emphasis is placed on shunt obstruction that the reader may assume that the experiments involve astrocytes attached to silicone catheters. More importantly, experimental design details are missing, i.e. duration of shear stress and the procedure for producing shear stress.
 - ✓ The Abstract text has been tightened to the recommended length of approximately 150 words. The study is directly applicable to silicone catheters because of the use of a silicone surface and physiologically relevant shear in our experimentation; still, we have tailored this text accordingly and appreciate this feedback.
The time interval for real-time cytokine secretion imaging under shear stress has been added to the Abstract (Line 25). The microfluidic shear device chip for producing shear stress has been added to the Abstract (Line 22).

Lines 16-27: “It has been hypothesized that physiological shear forces acting on medical devices implanted in the brain significantly accelerate the rate to device failure in patients with chronically indwelling neuroprosthetics. In hydrocephalus shunt devices, shear forces arise from cerebrospinal fluid flow. The shunt’s unacceptably high failure rate is mostly due to obstruction with adherent inflammatory cells. Astrocytes are the dominant cell type bound directly to obstructing shunts, rapidly manipulating their activation via shear stress-dependent cytokine secretion. Here we developed a total internal reflection fluorescence microscopy combined with a microfluidic shear device chip for quantitative analysis and direct spatial-temporal mapping of secreted cytokines at the single-cell level under physiological shear stress to identify the root cause for shunt failure. Real-time secretion imaging at 1-min time intervals enabled successful detection of a significant increase of astrocyte IL-6 cytokine secretion under shear stress greater than 0.5 dyne/cm², validating our hypothesis and highlighting the importance of reducing shear stress activation of cells.”

Introduction

3. Overall, this Introduction is very informative but excessively long and filled with details that should be reserved for the Discussion (e.g. lines 70-115). As a result, it takes forever to get to the main objectives of the study, which are partly technical and partly neurobiological.

✓ We thank the reviewer for pointing this out. The Introduction text has been tightened by adding lines 70-87 to the Discussion (Lines 358-370). Lines 88-115 are retained in the Introduction as the background, rationale, major results and conclusions for the work (Lines 69-95).

4. The writing style should be revised in many instances. Some examples are:

(a) in line 64, the word “exasperated” should be replaced by a less-emotional word such as “heightened”;

✓ The word “exasperated” has been replaced by “heightened”:

Line 62: “...heightened activation and proliferation of other astrocytes...”.

(b) in line 83, “Emphasizing the dominant role of cytokines.... is not a sentence;

✓ “Emphasizing the dominant role of cytokines....” has been made into a sentence:

Lines 57-60: “Cytokine secretion creates spatially and temporally varying concentration profiles in the extracellular environment to communicate, activate and recruit other inflammatory cells, emphasizing the dominant role of cytokines in orchestrating the dynamic crosstalk among cells.”

(c) in lines 90-91, "...intensive wash steps, 91 which causes a lag...;

✓ The word "lag" has been replaced by "delay":

Lines 72: "...several inherent drawbacks such as intensive wash steps, which causes a delay between secretion and detection."

(d) in lines 92-93, "...observation of a cellular variables..."

✓ An example has been added for "a cellular variable" to make the sentence clearer:

Lines 74-75: "...simultaneous real-time observation of a second cellular variable (e.g. cell physiological states, intercellular adhesion molecule expression) at the time of cytokine secretion."

5. In line 41, "triggers" should be plural.

✓ The word "triggers" has been replaced by "trigger":

Lines 30-31: "Persistent motion-related shear forces of medical devices implanted in the brain trigger a rather complex cascade of foreign body reactions (FBR)..."

6. In line 50, shear stress may not be the only important driver of scar formation, so "the key" should be softened.

✓ The word "the key" has been replaced by "a key":

Lines 43-44: "This fact increases the shear stress at the proximal segment and is a key driver of a dense glial scar formation around devices causing failure via obstruction."

Results

7. At times the text in this section often begins with excessive background information that does not belong in a Results section (e.g. lines 120-135) and should be moved to the Discussion section.

✓ This observation is correct. The Results background information has been moved to the Discussion (Lines 370-373).

8. The statement in lines 154-156 should be referenced.

- ✓ The statement has been referenced:

Lines 125-127: “We know that the initial phase of the FBR is blood-device interactions, which occurs immediately upon implantation caused by vasculature or BBB disruption, resulting in the influx of serum proteins and their nonspecific adsorption to the device surface [33].”

[33] Karumbaiah, L. et al. Relationship between intracortical electrode design and chronic recording function. *Biomaterials* 34, 8061–8074 (2013).

9. Most importantly, how do we know these are (a) astrocytes and (b) that they are single cells? It is impossible to know from the photographs in Figure 4D, and in the Methods section that follows there is no description of how the cells are obtained/harvested and identified. This is important because the statement is made that “Dynamic cellular states for live single cells could also be observed such as a change in astrocyte phenotype from a resting to a reactive form under shear-cytokine activation, characterized by horizontally swollen bodies.”; other features of resting vs reactive astrocytes, i.e. branching patterns, have not been evaluated, and the astrocyte “swelling” has not been quantified.

We believe the observation of the reviewer is exact. We have revised the text to address the concerns and hope that it is now clearer.

- (a) In Figure 4D, we know the cells are astrocytes since human astrocytes were purchased from ScienCell Research Laboratories (Catalog #1800) commercial cell lines (see the Methods section). Our lab has tested these cells extensively (over low and high passage, varying protein content and media type, and flow rates) far passed the scope of this work for expression of glial fibrillary acidic protein (GFAP), a protein highly specific for astrocytes. Within all our work, we have found 100% of cells expressing GFAP. For improved clarity the following sentence has been added to the Methods:

Lines 434-435 and 453-454: “Human astrocyte cells were purchased from ScienCell Research Laboratories (Catalog #1800) commercial cell lines. Astrocytes were tested for presence of glial fibrillary acidic protein, of which all cells are positive in the culture conditions described.”

- (b) Single astrocyte cells were identified and localized in individual image frames at least 100 μm apart from other astrocytes. Based on single-cell studies, the distance over which cytokine-mediated communication happens effectively is far less than 100 μm . Therefore, since cytokine signaling are limited to small distances of only a few cell diameters, substantiates that the IL-6 cytokine secretion detected is only from the identified single astrocyte cell [44, 45, 46]. For improved clarity the following sentence has been added to the Discussion:

Lines 291-295: “Single astrocyte cells were identified and localized in individual image frames at least 100 μm apart from other astrocytes. Since the distance over which cytokine-mediated communication happens effectively is far less than 100 μm , and cytokine signaling are limited to small distances of only a few cell diameters [44, 45, 46], substantiates that the IL-6 secretion detected is only from the identified single astrocyte cell.”

[44] Oyler-yaniv, A. et al. A Tunable Diffusion-Consumption Mechanism of Cytokine Propagation Enables Plasticity in Cell-to- Cell Communication in the Immune System. *Immunity* 46, 609–620 (2017).

[45] Bagnall, J. et al. Quantitative analysis of competitive cytokine signaling predicts tissue thresholds for the propagation of macrophage activation. *Sci. Signal.* 11, 1–16 (2018).

[46] Thurley, K., Gerecht, D., Friedmann, E. & Höfer, T. Three-Dimensional Gradients of Cytokine Signaling between T Cells. *PLOS Comput. Biol.* 1–22 (2015). doi:10.1371/journal.pcbi.1004206

In Figure 4D, swelling of reactive astrocytes was quantified as the increase in cross-sectional area relative to the resting astrocytes. The cross-sectional area was calculated using a line drawn to measure the diameter of astrocyte boundary [28, 34]. Future work will expand on the degree of cell reactivity by quantifying astrocyte processes directionality and bifurcation number. For improved clarity the following sentence has been added to the Results and Discussion:

Lines 197-198 and 222-223 and 296-297: “Swelling of reactive astrocytes was quantified as the increase in cross-sectional area relative to the resting astrocytes [28, 34]. The cross-sectional area was calculated using a line drawn to measure the diameter of astrocyte boundary. Future work will expand on the degree of cell reactivity by quantifying astrocyte processes directionality and bifurcation number.”

[28] Liddelov, S. A. et al. Neurotoxic reactive astrocytes are induced by activated microglia. *Nature* (2017). doi:10.1038/nature21029

[34] Edema, U. C. et al. The Cellular Mechanisms of Neuronal Swelling Underlying Cytotoxic Edema. *Cell* 161, 610–621 (2015).

The statement made that “Dynamic cellular states for live single cells could also be observed such as a change in astrocyte phenotype from a resting to a reactive form under shear-cytokine activation, characterized by horizontally swollen bodies.” is to emphasis that the TIR-FM/MSDC assay platform provides a powerful tool for analyzing molecule secretion dynamics with simultaneous monitoring of cellular physiological states by live-cell imaging. For improved clarity the sentence has been modified:

Lines 194-196: “Via this unique assay platform, simultaneous dynamic cellular states for live single cells could also be observed such as a change in astrocyte phenotype from a resting to a reactive form under shear-cytokine activation, characterized by horizontally swollen bodies.”

10. Important portions of Figures 1 and 2 are extremely small and difficult to read (e.g. Figure 4c&d).

✓ The text in Figures 1, 2, and 4c&d were enlarged for reading enhancement.

11. Validations of the TIR-FM/MSDC biosensor (Figure 3) are excellent, with caveats for the semi-quantitative nature of the data.

✓ The capture ratio of the TIR-FM/MSDC biosensor depends upon variables such as the height of the cytokine release point (which determines the probability of the cytokine encountering the capture antibody). Therefore, the assay platform developed in this study is proficient for in situ detection of the onset of cytokine secretion from single cells at high temporal resolution in a sensitive and selective manner, while also providing semi-quantitative data on secreted cytokines. For improved clarity the sentence has been modified:

Lines 159-164: “The capture ratio of the TIR-FM/MSDC biosensor depends upon variables such as the height of the cytokine release point (which determines the probability of the cytokine encountering the capture antibody). Therefore, the assay platform developed in this study is proficient for in situ detection of the onset of cytokine secretion from single cells at high temporal resolution in a sensitive and selective manner, while also providing semi-quantitative data on secreted cytokines (Fig. 3b).”

Methods

12. Although astrocyte-promoting media are used, no descriptions of the source of astrocytes has been provided; thus, it is not known if they are commercial cell lines or cells derived from primary harvesting and cell culture.

✓ Human astrocytes were purchased from ScienCell Research Laboratories (Catalog #1800) commercial cell lines. For improved clarity the following sentence has been added to the Methods:

Lines 434-435: “Human astrocytes were purchased from ScienCell Research Laboratories Catalog #1800) commercial cell lines.”

13. Likewise, no cell labeling has been provided to confirm that these cells are astrocytes. This problem is compounded because the magnification used in Figure 4 does not permit visualization of the branching pattern of the cells, which could be used to distinguish astrocytes.

- ✓ We have acknowledged this fact and modified the text accordingly. Please refer to question 9(a) above.
TIR-FM imaging was performed using an Apochromat 100× objective, therefore the single astrocyte cells were identified to monitor spatial-temporal mapping of secreted cytokines. Although visualization of the branching pattern of the astrocyte cells would be appealing, considering that only commercial astrocyte cell lines were used, no cell labeling was required to distinguish astrocytes.

14. On line 479, the Author Instructions call for this heading to be “titled “Statistics and Reproducibility””.

- ✓ The title heading has been replaced by “Statistics and Reproducibility”.

Line 502: “Statistics and Reproducibility.”

Discussion

15. The point about IL-6 activation of astrocyte proliferation is very well taken (and should be referenced), but some mention should be made of the caveat that these relatively short-term experiments could not determine if astrocyte proliferation occurred in this in vitro system.

- ✓ The following references have been added to line 277:

[39] Zamanian, J. L. et al. Genomic Analysis of Reactive Astroglia. *J. Neurosci.* 32, 6391–6410 (2012).

[40] Selmaj, K. W., Farooq, M., Norton, W. T. & Raine, C. S. Proliferation of astrocytes in vitro in response to cytokines. A primary role for tumor necrosis factor. *J. Immunol.* 144, 129–135 (1990).

- ✓ The following sentence has been added to the Discussion:

Lines 277-280: “Moreover, recent long-term in vivo data collected in our lab indicate astrocyte markers in obstructive masses on shunts to be co-localized with proliferative markers, indicating that astrocytes are active on the shunt surface: they produce inflammatory cytokine IL-6 and proliferate.”

16. The two sentences in lines 305-308 are awkward; “In contrast to the general implant failure...” does not have a verb and thus is not a complete sentence.

✓ The sentence has been revised into a complete sentence:

Lines 282-285: “This is strongly in accordance with other reports, which indicate a necessity for control of the degree of inflammatory cell activation for considerable improvement of device performance within the brain, in contrast to the general implant failure of reduced cell adhesion on the device surface in vivo.”

17. Likewise, “Possibly, due to cell spreading...” (line 335) is not a complete sentence.

✓ The sentence has been revised into a complete sentence:

Lines 317-320: “Under conditions of high shear stress, astrocytes on hydrophobic PDMS substrates resulted in higher IL-6 secretion compared to hydrophilic glass substrates, possibly due to cell spreading and contact between the attached cells and ligand adsorption on different surfaces.”

18. The statements made in lines 352-355 on the effects of IL-6 secretion seem contradictory and should be clarified.

✓ The statement has been revised for improved clarity:

Lines 338-339: “ELISpot data confirm the results obtained from the TIR-FM/MSDC biosensor.”

19. In line 382, what does “density” mean?

✓ Based on the Bernoulli's equation, since density and pressure differences do not change between the two shunt designs, velocity will also be the same for both. But since the Bernoulli's equation has not been mentioned in the manuscript the sentence has been revised as follows:

Lines 387-391: “However, shear stress will be higher in the proximal holes of these shunts. Based on fluid shear stress equation of $\tau = \mu \, du/dy$, where μ is dynamic viscosity, and du/dy is the gradient of velocity in the direction perpendicular to the flow, since the gradient viscosity of the decreasing hole diameters is higher, shear stress will be higher for the proximal holes.”

20. The important statement about drug interventions in lines 398-399 should be referenced. In fact, it would be worthwhile to mention some of the experimental studies that have attempted to reduce neuroinflammation and astrogliosis.

- ✓ The statement about drug interventions has been referenced ([65], Line 410), and the following sentence has been added to the Discussion:

Lines 418-421: “For example, since IL-6 production is considered a downstream sequence related to TNF- α /IL-1 β stimulation of cells, FDA-approved neutralizing monoclonal antibody therapies that inhibit human TNF- α , IL-1 β , and the IL-1 receptor have been developed to decrease their activity [28, 65, 67].”

[28] Liddelov, S. A. et al. Neurotoxic reactive astrocytes are induced by activated microglia. *Nature* (2017). doi:10.1038/nature21029

[65] Karimy, J. K. et al. Inflammation in acquired hydrocephalus: pathogenic mechanisms and therapeutic targets. *Nat. Rev. Neurol.* 16, 285–296 (2020).

[67] Dinarello, C. A., Simon, A. & Van Der Meer, J. W. M. Treating inflammation by blocking interleukin-1 in a broad spectrum of diseases. *Nat. Rev. Drug Discov.* 11, 633–652 (2012).

21. It might be worth mentioning the “bystander effects” of increased CSF cytokine levels on the brain parenchyma, especially the periventricular white matter where neuroinflammation is known to occur.

- ✓ The following sentence has been added to the Discussion:

Lines 373-379: “IL-6 promotes neuronal survival and neurite growth. These features are indications of the beneficial roles of IL-6 in repair and modulation of inflammation in the CNS. However, the overproduction of IL-6 is associated with glial scar formation, and periventricular white matter (where neuroinflammation occurs) injury due to increased CSF concentrations of IL-6 [59]. Thus, a careful inflammatory balance of IL-6 is necessary for proper repair. Inhibition of both IL-6 and IL-6r by antibody neutralization reduces damage to surrounding brain parenchyma as a result of bystander effects of increased CSF cytokine levels [60].”

[59] Habiyaremye, G. et al. Chemokine and cytokine levels in the lumbar cerebrospinal fluid of preterm infants with post-hemorrhagic hydrocephalus. *Fluids Barriers CNS* 14, 1–10 (2017).

[60] Ramesh, G., Maclean, A. G. & Philipp, M. T. Cytokines and chemokines at the crossroads of neuroinflammation, neurodegeneration, and neuropathic pain. *Mediators Inflamm.* 2013, (2013)

Comments of Reviewer # 2 and response by the authors:

Summary: Shunt malfunction is a common problem in patients with hydrocephalus, with occlusion representing the most common cause. Here, the authors hypothesize that the effects of shear stresses at the proximal end of the shunt catheter are strongly responsible for astrocyte activation via the secretion of the pro-inflammatory cytokine, IL-6. The authors combine a microfluidic chip to simulate shear forces and surface characteristics with fluorescence-based imaging and finite element models to interrogate the role of shear-stresses in IL-6 secretion. Overall, the study is clear, convincing and well-written. It represents an important contribution to the fields of both basic biology and device design. I recommend publication with minor revisions, with specific questions outlined below.

Thank you so much for the positive evaluation of our manuscript. We greatly appreciate the thorough and thoughtful comments provided. We made sure that each one of the comments has been addressed carefully and the paper is revised accordingly.

1. In the study, 4 different microchannels with varying diameters simulate shear stresses in a high throughput manner. As a confirmation of the hypothesis, it would be useful to demonstrate the effects of microchannels with varying diameters along their length (i.e, narrowing at various points along their length) to induce shear stress concentrations.
 - ✓ In this study the focus is on the effects of physiological shear stress at the proximal end of the hydrocephalus shunt providing valuable information to identify the root cause for shunt failure. Computational fluid dynamics simulations have shown that in CSF shunts, the wall shear stress at the proximal holes is greater than 0.5 dyne/cm^2 . Thus, in this study high uniform wall shear stress of more than 0.5 dyne/cm^2 across the microchannel's surface is maintained to examine the correlations between physiological shear stress applied on single astrocytes and the resulting IL-6 cytokine secretion. Therefore, the answer to our question of whether shear stress significantly accelerates astrocyte activation on the CSF shunt surface is confirmed. However, it would be quite interesting to see how potential flow and shear change as microchannel diameter changes such as blood vessels. This could be done in future work. The following sentence has been added to the Discussion lines 422-424.

“For future work, it would be quite interesting to demonstrate the effects of a microfluidic shear device chip composed of a series of microchannels with varying diameter along their length (i.e., narrowing at various points along their length) to induce fluid flow shear stress [68].”

[68] Feng, S., Mao, S., Zhang, Q., Li, W. & Lin, J. M. Online analysis of drug toxicity to cells with shear stress on an integrated microfluidic chip. *ACS Sensors* 4, 521–527 (2019).

- ✓ Also, data in our lab indicate that astrocytes have an increased attachment propensity in vitro with increasing flow-induced shear stress (shear stress from 0.039 to 0.134 dyne/cm²) [10, 15, 16]. Introduction lines 51-52.
2. While the authors make a convincing argument that reducing shear-stresses at the proximal end of the shunt would reduce fibrosis and occlusion, there are practical aspects to shunt design, such as sizes when dealing with pediatric populations. These are particularly relevant when considering that shunt failure rates are highest in infants and younger patients. The study would be boosted if the author's discussed the implications of their findings in terms of sizing of shunt size and geometry, and their compatibility with pediatric populations.
- ✓ The reviewer brings up an excellent point. Another aspect to address is shunt size and geometry as it is relevant to the physical limitations of shunt design. Physical and surgical limitations of ventricular size, ventricular catheter environment, and ventricular catheter tip location to avoid hitting capillaries all have effect on the persistent high shunt failure rates in pediatric patients. As we continue to design shunts, we will be working with neurosurgeons to accentuate lower shear stress according to the physical limitations of ventricles. For example, exploring novel ways to improve the size of the holes and inner diameter for better compatibility with pediatric populations.

Examples of physical and surgical limitations are:

Decreasing the shunt size for pediatric populations may result in less contact between the walls and the shunt, resulting in less obstruction of the shunt inlets and less blockage of the CSF flow. However, as the child grows, migration of the shunt distal components gradually pulls the intraventricular catheter out of the ventricles and into the brain parenchyma, causing obstruction and shunt failure which requires surgical replacement of the ventricular shunt.

Infection is the second most common cause of ventricular shunt malfunction, which is more common in children. Therefore, high shunt failure rates in infants and younger patients might not only be the result of shunt size and geometry, but of infection.

References:

Paff, M., Alexandru-Abrams, D., Muhonen, M. & Loudon, W. Ventriculoperitoneal shunt complications: A review. *Interdisciplinary Neurosurgery* 13, 66-70 (2018).

Gmeiner, M., Wagner, H., Zacherl, C. et al. Long-term mortality rates in pediatric hydrocephalus—a retrospective single-center study. *Childs Nerv Syst* 33, 101–109 (2017).

- ✓ The following sentence has been added to the Discussion lines 395-401.

“Another aspect to address is shunt size and geometry as it is relevant to the physical limitations of shunt design. Physical and surgical limitations of ventricular size, ventricular catheter environment, and ventricular catheter tip location to avoid hitting capillaries all have effect on the persistent high shunt failure rates particularly when dealing with pediatric populations. As we continue to design shunts, we will be working with neurosurgeons to accentuate lower shear stress according to the physical limitations of ventricles. For example, exploring novel ways to improve the size of the holes and inner diameter for better compatibility with pediatric patients.”

3. CSF flow is highly intermittent. How does pulsing the flow rate affect IL-6 activation?

- ✓ Based on literature pulsing the flow rate governs the viscoelastic properties of the astrocyte cytoskeleton as stated in the referenced report: “Fluid shear generated non-uniform forces in actinin that critically depended on the stimulus rise time emphasizing the presence of viscoelasticity in the activating sequence”. Therefore, for a comprehensive investigation of pulsing flow rate effects on IL-6 activation, simultaneous measurements of cytoskeletal forces using a force sensitive actinin optical sensor is required at the time of cytokine secretion.

Reference:

Maneshi MM, Sachs F, Hua SZ. Heterogeneous Cytoskeletal Force Distribution Delineates the Onset Ca^{2+} Influx Under Fluid Shear Stress in Astrocytes. *Front Cell Neurosci.* 12 (2018)

- ✓ Furthermore, data in our lab indicate that for a 20-hour period, astrocyte adhesion was significantly increased under conditions of increased flow rate (0.25 and 0.30 mL/min). However, pulsation rate does not influence overall cell attachment in an acute, 20-hour period [15]. Whether these relationships would change over longer periods of time requires a suitable platform.
- ✓ The following sentence has been added to the Discussion lines 424-428.

“To maintain the environmental conditions observed in hydrocephalus, astrocyte activation under experimentally manipulated conditions of CSF pressure, pulsation rate, and flow rates over long periods of time serve as a potential area for future work. However, pulsation rate does not influence overall cell attachment in an acute, 20-hour period [15].”

[15] Harris, C. A. et al. Mechanical contributions to astrocyte adhesion using a novel in vitro model of catheter obstruction. *Exp. Neurol.* 222, 204–210 (2010).